# Multispectral and Thermal Sensors Onboard UAVs for Heterogeneity in Merlot Vineyard Detection: Contribution to Zoning Maps

**Luz K. Atencia Payares** [1,2,3,]*, **Ana M. Tarquis** [1,4], **Roberto Hermoso Peralo** [3], **Jesús Cano** [3], **Joaquín Cámara** [5], **Juan Nowack** [1,2] and **María Gómez del Campo** [1,2]

1. CEIGRAM, ETSIAAB, Universidad Politécnica de Madrid (UPM), Senda del Rey St. 13, 28040 Madrid, Spain; anamaria.tarquis@upm.es (A.M.T.); j.nowack@hotmail.com (J.N.); maria.gomezdelcampo@upm.es (M.G.d.C.)
2. Departamento deo Producción Agraria, ETSIAAB, Universidad Politécnica de Madrid (UPM), Ciudad Universitaria s.n., 28040 Madrid, Spain
3. Unmanned Technical Works (UTW), Leganés, 28919 Madrid, Spain; roberto.hermoso@utw.es (R.H.P.); jesus.cano@utw.es (J.C.)
4. Grupo de Sistemas Complejos, ETSIAAB, Universidad Politécnica de Madrid (UPM), Ciudad Universitaria s.n., 28040 Madrid, Spain
5. Diagnoterra, SL., 28029 Madrid, Spain; joaquin@diagnoterra.com
* Correspondence: lkatenciapayares@gmail.com

**Abstract:** This work evaluated the ability of UAVs to detect field heterogeneity and their influences on vineyard development in Yepes (Spain). Under deficit irrigation, vine growth and yield variability are influenced by soil characteristics such as water holding capacity (WHC). Over two irrigation seasons (2021–2022), several vegetation indices (VIs) and parameters of vegetative growth and yield were evaluated in two field zones. Multispectral and thermal information was obtained from bare soils. The water availability showed annual differences; it was reduced by 49% in 2022 compared to 2021, suggesting that no significant differences were found for the parameters studied. The zone with higher WHC also had the higher vegetative growth and yield in 2021. This agreed with the significant differences among the VIs evaluated, especially the ratio vegetation index (RVI). Soil multispectral and thermal bands showed significant differences between zones in both years. This indicated that the soil spectral and thermal characteristics could provide more reliable information for zoning than vine vegetation itself, as they were less influenced by climatic conditions between years. Consequently, UAVs proved to be valuable for assessing spatial and temporal heterogeneity in the monitoring of vineyards. Soil spectral and thermal information will be essential for zoning applications due to its consistency across different years, enhancing vineyard management practices.

**Keywords:** unmanned aerial vehicle (UAV); vineyards; VIs; spatial variability; zoning

## 1. Introduction

Viticulture increasingly requires information to optimise agricultural activities based on the variability of vineyards. Variations in environmental factors, topography, and characteristics of soils influence vine development [1]. Knowledge of this variability is necessary to manage each homogeneous unit independently, providing optimal inputs to each one. In this way, the highest grape production and quality at the lowest cost can be obtained [2].

Soil properties are one of the most critical parameters that determine vineyard development variability. The soil provides the vine with structural support, nutrients, and water. Some crucial aspects of vine development depend on the capacity of the soil to provide them, affecting physiology, production, and grape ripening dynamics [3].

In the central region of Spain, where irrigation limitations can be an issue, soil properties are the primary source from which grapevines extract water. Soil texture influences

soil water holding capacity (WHC) and, consequently, the progressive water release for vine root uptake [4]. This influence is even more decisive under deficit irrigation, a practice commonly applied in the study area. In soils characterised by low WHC, irrigation is a crucial element in mitigating the adverse effects of climate change [5].

Correct soil zoning is essential to manage a crop according to its corresponding zone's productive capacity and limitations [6]. In this way, growers can apply site-specific management strategies instead of implementing the same management practice throughout a whole vineyard, especially for irrigation management [7].

Sensors onboard drones (UAVs) are one solution to assess spatial and temporal heterogeneity in the development of woody crops. In these cases, vegetation does not entirely cover the surface of the soil, and the resolution provided by satellites is currently not less than 1 m. Higher resolution is necessary for trellis crops such as vineyards [8]. The variability of the vines can be observed through multispectral images, which have shown a correlation with several plant biophysical parameters [9]. Vegetation indices (VIs) can enhance vegetation cover based on its spectral response through algebraic operations of spectral bands [10–12].

At the same time, multispectral information with high resolution allows soil characterisation. Soil, like vegetation, has a spectral signature. Several studies have evaluated the usefulness of land surface spectral variation for the spatial prediction of soil attributes at a regional scale [13,14]. Remotely sensed soil data using spectral and colour photographs have been helpful for creating different field-extent soil property maps using different prediction models [15]. Knowledge of the soil and its relationship with vine growth and vigour is crucial for crop management [16,17]. However, to obtain conventional soil mapping, it is necessary for an expert surveyor to conduct fieldwork using aerial photo interpretation, topography, and vegetation maps [18]. Information on the soil and vegetation from high-resolution images makes it possible to generate maps that can be used for crop zoning based on the performance of crops [15,19], and to determine homogeneous management zones in vineyards.

Previous research has explored the generation of soil map zoning in vineyards using various approaches. Most studies have considered multi-temporal contexts and phenological stages of grapevines [20]. In contrast, others have used NDVI maps [21] and advanced algorithms to extract pure canopy multispectral information using sensor onboard UAVs [22] or aerial platforms [23] to generate vigour maps. Certain studies have also examined the use of soil physical properties to identify zones with similar plant yields based on vegetation indices (VIs) and vine water status [4]. Other studies have combined field-scale apparent soil electrical conductivity (ECap) and NDVI maps to select different fertility zones within vineyards [24,25].

Although these studies have commonly relied on multispectral data from plants to create homogeneous management zones, some have incorporated information on soil properties through field samples or nearby sensors. However, the potential utility of thermal and multispectral data from soils in determining differentiated zones within the vineyard has not yet been fully explored.

This study aimed to evaluate the potential of UAV sensors for detecting spatial and temporal heterogeneity in vineyards. This research assessed vine development in soils with varying hydraulic characteristics under the same management practices by comparing data collected from an UAV and conventional field techniques over different climatic years. Understanding the capabilities of UAVs to distinguish vine responses and soil types could provide valuable insights for creating zoning maps and improving vineyard management practices.

## 2. Materials and Methods

The flowchart in Figure 1 shows the organisation of the data obtained in this study. The data collected from soil and vines in the field were compared to the data obtained from

an UAV. The objective was to evaluate the UAV's sensitivity to soil heterogeneity and vine response.

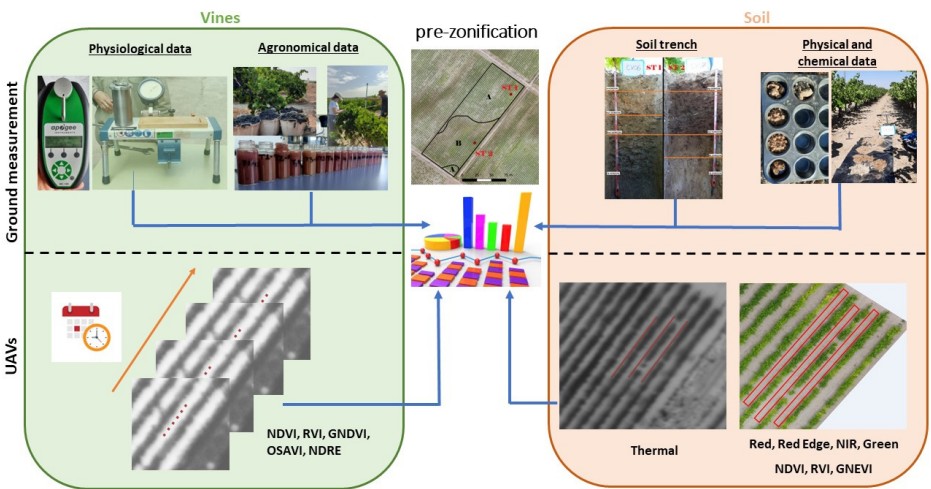

**Figure 1.** Flowchart of the different sources of information used to describe the spatial and temporal heterogeneity in a commercial vineyard in Yepes-Toledo.

### 2.1. Site and Vineyards Description

The commercial vineyards were in Yepes, Toledo (Spain), located at 39°56′25.8″ N 3°43′22.4″ W (WGS84, UTM zone 30 N), at an altitude of 570 masl, with an extension of around 40 ha. An experimental zone of 13,900 m$^2$ was selected for this study.

Twenty-year-old Merlot grapevines (Vitis vinifera L.), grafted on SO4 (Selection Oppenheim 4) rootstock, are grown in the selected area. Vine rows are oriented northeast and arranged on a trellis, with a plantation frame of 2.60 × 1.10 m. The study was conducted during the 2021 and 2022 irrigation seasons.

A weather station in Magán provided temperature, rainfall, and ETo data (Toledo, 505 masl, latitude 40°2′5.81″ N and longitude 3°20′3.49″ W, Huso UTM30 Coord) (Figure 2).

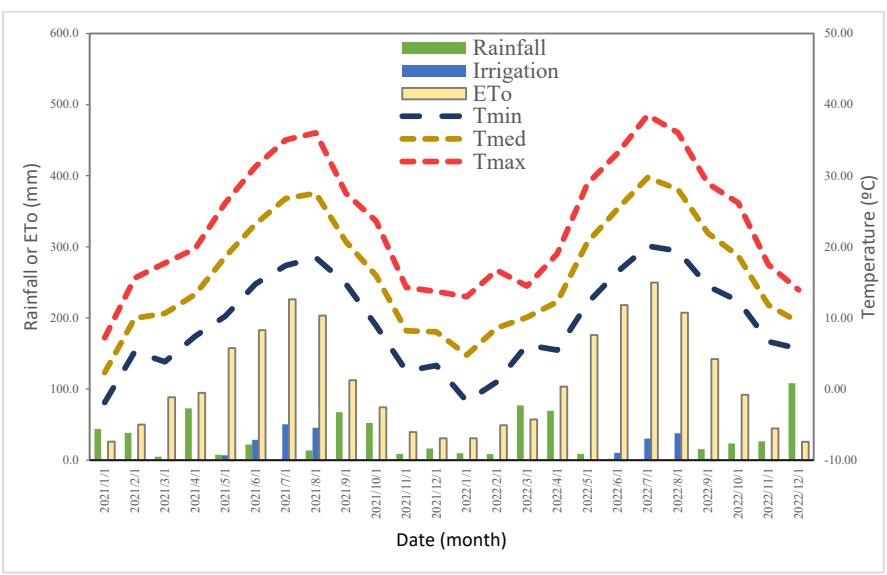

**Figure 2.** Monthly rainfall, reference evapotranspiration, and minimum and maximum temperatures from October 2020 until September 2022 located in the experimental vineyard (Toledo, Spain).

The site's climate is Mediterranean, with warm, dry summers and mild winters. The average daily temperature is 15.3 °C. The area is characterised by low average rainfall (338.4 mm) and high evaporative demand (1350 mm).

Irrigation was applied with one drip emitter for each meter with an application rate of 2l h$^{-1}$. It was scheduled according to the standard practices followed by the Bodegas Casa del Valle, i.e., with limited time intervals. The irrigation started on 17 May 2021 and finished on 27 August 2021. In 2022, the irrigation started on 23 June 2022 and finished on 01 September 2022. The amounts of water applied during the irrigation season was 131 and 79 mm in 2021 and 2022, respectively.

### 2.2. Preliminary Zoning Map and Soil Properties

A photo interpretation was applied based on historical vineyard maps over 30 years of evolution of the National Aerial Orthophotography Plan. Different homogeneous zones were identified.

The zoning was carried out by considering two criteria: (i) the soil colour difference and (ii) the zones with different colours having a representative area (higher than 100 m$^2$). The criteria both ensured adequate simplicity and representativeness of the area (Figure 3a).

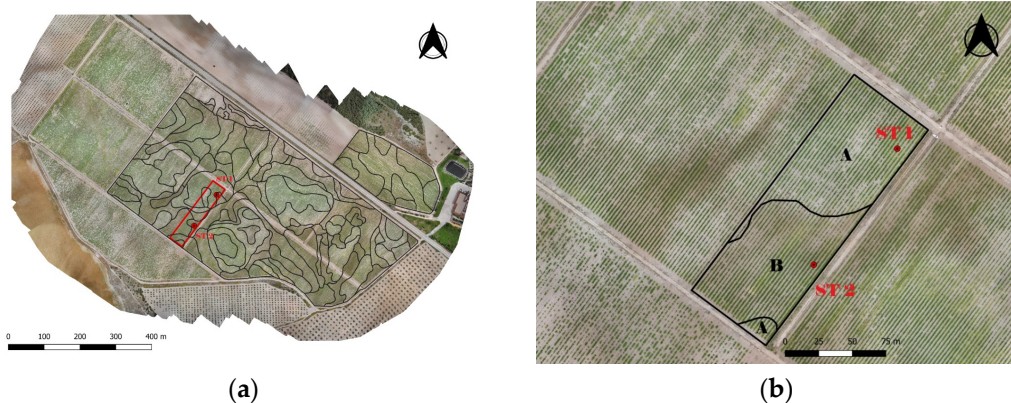

(**a**) (**b**)

**Figure 3.** (**a**) Aerial photograph of the experimental vineyard. Lines indicate the preliminary soil zoning of the commercial vineyard, the red rectangle indicates the experimental zone, the locations of the two soil trenches (ST1 and ST2) are indicated with red points; (**b**) soil map details corresponding to the experimental zone's final zoning in the vineyard (Toledo, Spain).

The final soil map of the vineyard plots was obtained following the standards of Order 1 of Soil Surveys included in the USDA *Soil Survey Manual* [26], with 0.5 soil observations per hectare among soil trench and manual auger probes. Soil map units were delineated by aerial photo interpretation of aerial photograms and orthophotos. The spatial resolution was 0.25 metres/pixel using geomorphological criteria, mainly relief and lithology, according to photographic patterns associated with defined textures and tones. Soil boundary lines were fitted using remote sensing data and the digital terrain model (2 m pixel size). Once the zones were delimited (black box) (Figure 3b), in a 13,900 m$^2$ experimental zone, two soil trenches (ST1 and ST2) were dug at representative points.

Properties of the horizons of two different soils in one trench each (ST1 and ST2) were described following soil surveys included in the USDA *Soil Survey Manual* [25], which allowed the soil to be classified [27]. Several soil properties for each horizon were analysed: texture, depth, organic matter (OM), electrical conductivity from the saturated paste extracts (1:2.5 ratio) (ECe), pH, active limestone, and total nitrogen. For these measurements, three repetitions of each sample were analysed. The available water (AW, mm) and soil WHC (mm) were calculated as follows:

$$AW = Hd \times (aw/100) \times BD \times 1000 \tag{1}$$

$$WHC = AW \times ((100 - GV)/100) \tag{2}$$

where aw is the available water (% weight) obtained as differences between water content at −0.33 bar (field capacity) and −15 bar (wilting point), BD is the bulk density (t/m³), Hd is the horizons depth (m), and GV is the gravel (particle size greater than 2 mm) (% vol).

## 2.3. Physiological and Agronomical Parameters

The main phenological stages (budburst, flowering, veraison, and harvest-ripe stages) were defined using the phenological scale of Eichhorn and Lorenz (1977), modified by Coombe (1995) [28]. The physiological and agronomical parameters were measured in 12 experimental vines established in the study soils (ST1 and ST2), with six plants per soil trench.

Physiological parameters such as the stem water potential (SWP) and chlorophyll content (Chl) were measured in the 12 experimental vines at 9 and 12 h solar time for 9 days (25 June 2021, 5 July 2021, 20 July 2021, 30 July 2021, 18 August 2021, 30 June 2022, 15 July 2022, 5 August 2022, and 12 August 2022). SWP was assessed in healthy mature shaded leaves enclosed for 1 h in aluminium foil bags to reach the water status equilibrium between leaf and stem. SWP was measured using a Scholander pressure chamber (Soil Moisture Equipment Corp., Santa Barbara, CA, USA).

Chl was simultaneously measured in three leaves per each experimental vine using an Apogee MC-100 instrument (Apogee Instruments, Inc., Logan, UT, USA) on the same days and times as the SWP assessment.

The canopy was described for 12 experimental vines on 28 June 2021 and 20 June 2022. A flexible tape measured the total canopy contour at three different points of each vine: the trunk and 40 cm apart to both sides. The positions of the highest and lowest leaves were noted at the same points. The canopy width was measured at each point at three different heights (80, 110, and 120 cm). These data were used to calculate the external surface area (ESA) and canopy volume. Leaf density was estimated as a canopy gap. A vine canopy photo of each experimental vine with a red blanket in the back was taken to delimit it (Figure 4). Using ImageJ software (Wayne Rasband., Bethesda, MD, USA), the photos were binarized to calculate the gaps in the canopy.

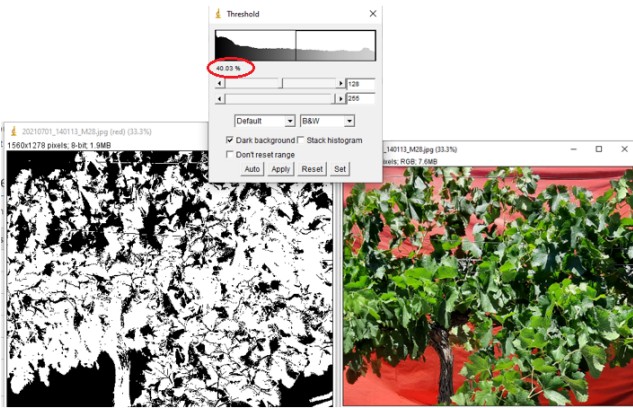

**Figure 4.** Screenshot of the ImageJ program used to obtain the percentage value of the gaps of the vines by transforming the RGB images.

The experimental vines were harvested on 20 August 2021 and 16 August 2022, production was weighed, and bunches were counted. One hundred berries per experimental vine were sampled at harvest. Samples were put into tagged plastic bags, placed in a portable cooler with ice, and taken to the laboratory. They were immediately weighed and processed to determine total soluble solids (°Brix) using an Atago refractometer Brix digital (ATAGO CO., LTD., Tokyo, Japan) and total acidity using an automatic neutraliser 702 SM Titrino (Metrohm AG, Herisau, Switzerland) and a pH meter Hach sensION (Hach company, Loveland, CO, USA), according to Glories (2001) method. Therefore, the final values corresponded to the harvest date of each year.

*2.4. UAV Images*

The eBee SenseFly fixed-wing UAV platform (AgEagle Aerial Systems Inc. Wichita, Kansas), equipped with a Parrot Sequoia multispectral sensor (Parrot© SA, 2017, Paris, France) and a Duet-T sensor (AgEagle Aerial Systems Inc., Wichita, Kansas), was used to collect multispectral and thermal data. The UAV surveys were conducted by flying 120 m above ground level at 12 solar time to avoid the shadow effect. An average 0.148 m multispectral and 0.16 m thermic pixel−1 ground image resolution was obtained.

UAV flight parameters were as follows: speed of 50 km/h; overlap Duet-T sensor, lateral overlap of 80% and longitudinal overlap of 80%; overlap Multispectral sensor, lateral overlap of 70% and longitudinal overlap of 70%.

The multispectral sensor had four bands: green (530–570 nm), red (640–680 nm), red edge (730–740 nm), and near-infrared (NIR) (770–810 nm) bands. Before the flight, a dedicated Sequoia equipment calibration plate was recorded with the Parrot Sequoia camera to normalise the local lighting. The thermal sensor included a high-resolution thermal infrared camera and a senseFly SODA RGB camera. The images were recorded during clear sky conditions.

The multispectral sensor was used to obtain soil information on 17 May 2021 and 19 May in 2022. The thermal sensor was used to obtain soil information on 15 May and 20 July in 2021 and on 17 May and 15 July in 2022. The multispectral sensor was used to obtain canopy information on 25 June, 5 July, 20 July, 30 July, and 18 August in 2021 and 30 June, 15 July, 05 August, and 12 August in 2022.

The experimental vines were identified using QGIS 3.4 software (QGIS, Free Software Foundation, Boston, MA, USA). Each canopy vine was delimited, and 12 squares (one square for each plant) of 0.30 × 0.30 m were extracted (Figure 5a), avoiding the effects of shadows and soil components. The number of pixels obtained with the multispectral camera was 4–6 pixels/vine.

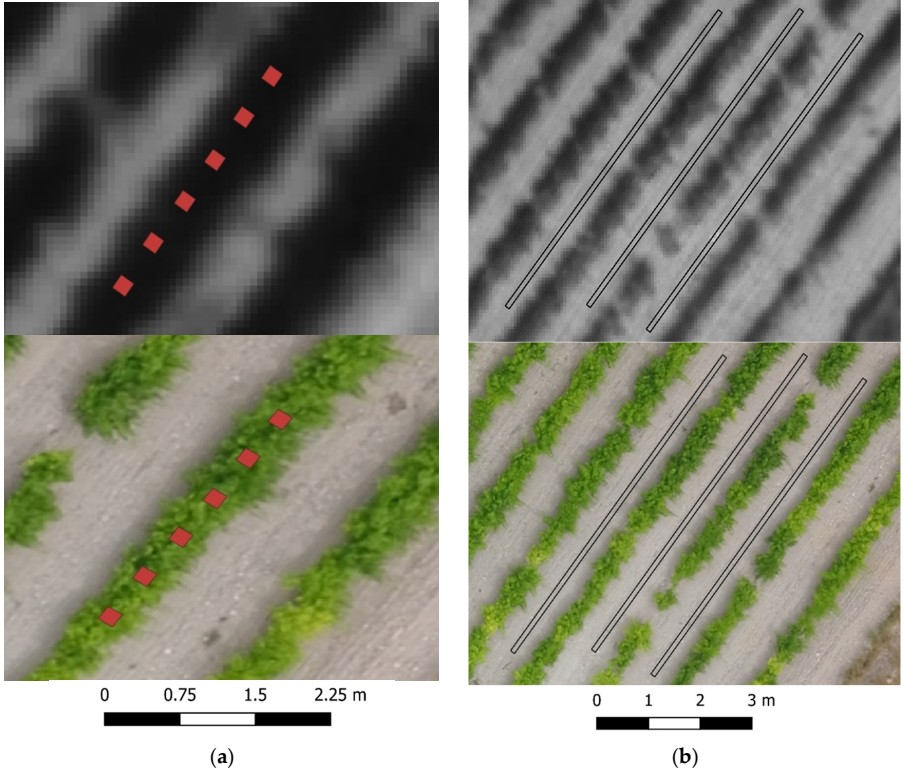

**Figure 5.** Multispectral image detail with the location of the canopy experimental vine (**a**) and soil (**b**) in one of the study soil plots (ST1).

The soil near the experimental vines was delimited in three rectangles of 0.19 × 15 m per ST (Figure 5b). The total number of pixels per soil was 520–530 pixels/ST for the multispectral sensor and 340–350 pixels/ST for the thermal sensor.

Vegetation Indices

The following VIs were calculated for the vine canopy and soil (NDVI, GNDVI, and RVI, calculated for the soil).

The normalised difference vegetation index (NDVI) is an index that quantifies the amount of vegetation in an area and its health [29]. It relates the reflected radiation in the red and near-infrared (NIR) bands of the electromagnetic spectrum. Its expression is as follows:

$$NDVI = (NIR - Red)/(NIR + Red) \tag{3}$$

Among the many available VIs, the NDVI is the most widely used [30].

The green normalized difference vegetation index (GNDVI) is computed similarly to the NDVI. The green band is used instead of the red band [31]. It is related to the proportion of photosynthetically absorbed radiation, and is linearly correlated with the leaf area index (LAI) and biomass [30]. Thus, the GNDVI is more sensitive to the chlorophyll concentration than the NDVI, and ranges from 0 to 1.0 [32]:

$$GNDVI = (NIR \times Green)/(NIR + Green) \tag{4}$$

The ratio vegetation index (RVI) indicates plant canopy vigour [33], based on the principle that leaves absorb relatively redder than infrared light, and is expressed as follows:

$$RVI = NIR/Red \tag{5}$$

The NDVI is not always the most accurate choice for detecting crop anomalies. This index does not consider the red edge band. This zone marks the limit between absorption by chlorophyll in the red band and scattering due to internal leaf structure in the NIR band [34]. The red edge position is susceptible to changes in vegetation properties, which researchers can easily exploit [30]. Similar to the NDVI, the normalised difference red edge (NDRE) is defined as:

$$NDRE = (NIR - RedEdge)/(NIR + RedEdge) \tag{6}$$

The optimised soil-adjusted vegetation index (OSAVI) uses a soil adjustment coefficient (0.16) as the optimal value to minimise variation with the soil background [35]. In terms of performance, it is similar to other indices of the SAVI class, and its chief advantages are its simplified formulation and the lack of a requirement for a priori knowledge of the soil type. The residual variation in the OSAVI is due to the soil being evenly spread across the full range (0–1) of the crop ground cover, making this index particularly suitable for agricultural applications [36]. It is expressed as:

$$OSAVI = (NIR - Red)/(NIR + Red + 0.16) \tag{7}$$

Thermal and multispectral data were subjected to analysis of variance (ANOVA) using Infostat version 1.5 (National University, Córdoba, Argentina). The means were separated using the LSD test (<0.05) for statistical differences. A Wilcoxon non-parametric test was used to assess differences between the mean physiological and agronomical data, using the R studio statistical software (RStudio Inc., Boston, MA, USA).

## 3. Results

### 3.1. Environmental Conditions

The years 2021 and 2022 were both characteristic of the Mediterranean area (Figure 2). August was the hottest month in 2021 (Tmean 27.5 °C), and July was the hottest month

in 2022 (Tmean 29.8 °C). January was the coldest (Tmean 2.3 and 4.7 °C in 2021 and 2022, respectively). The highest temperature (44 °C) was recorded on 14 August 2021, and the lowest temperature (−15 °C) was recorded on 12 January 2021.

Annual rainfall was similar between years (Figure 2). It rained 348.7 and 348.3 mm in 2021 and 2022, respectively. Rainfall occurred mainly in April, September, and October 2021. However, in 2022, most of the precipitation occurred in March and April. The 2022 autumn season was drier compared with the previous year. During the experimental period (May–August), the conditions were highly evaporative with high cumulative ETo (1284 and 1395 mm in 2021 and 2022, respectively), and available water was scarce (rain + irrigation were 175 and 89 mm in 2021 and 2022, respectively). The irrigation was applied from 17 May 2021 to 27 August 2021 and from 23 June 2022 to 1 September 2022. The amounts of water used during the irrigation season were 131 and 78 mm in 2021 and 2022, respectively. In 2022, there were 40 and 49% reductions in irrigation and rain + irrigation, respectively, compared to 2021, due to low water availability during this year. In both years, water provided by irrigation was insufficient to cover the high evaporative demands, and deficit irrigation was used.

### 3.2. Soil Zone Mapping

In the preliminary soil zoning performed in the commercial vineyard using photo interpretation, seven different soil types were observed in the experimental plot. This is delimited by the red box (Figure 3a). The information obtained from two soil trenches through the observation and analysis of samples from the different layers of the profiles (Figure 6) reduced the zoning classification to two monotaxic soil map units. Figure 3b shows the experimental zone plot with the final zoning lines. Soils in "A" correspond to coarse-loamy, gypsic, mesic Typic Calcixerepts, and soils in "B" correspond to coarse-loamy, mixed, active, mesic Calcic Haploxeralfs. The area of zone "A" was 7315 m$^2$ and the area of zone "B" was 6535 m$^2$.

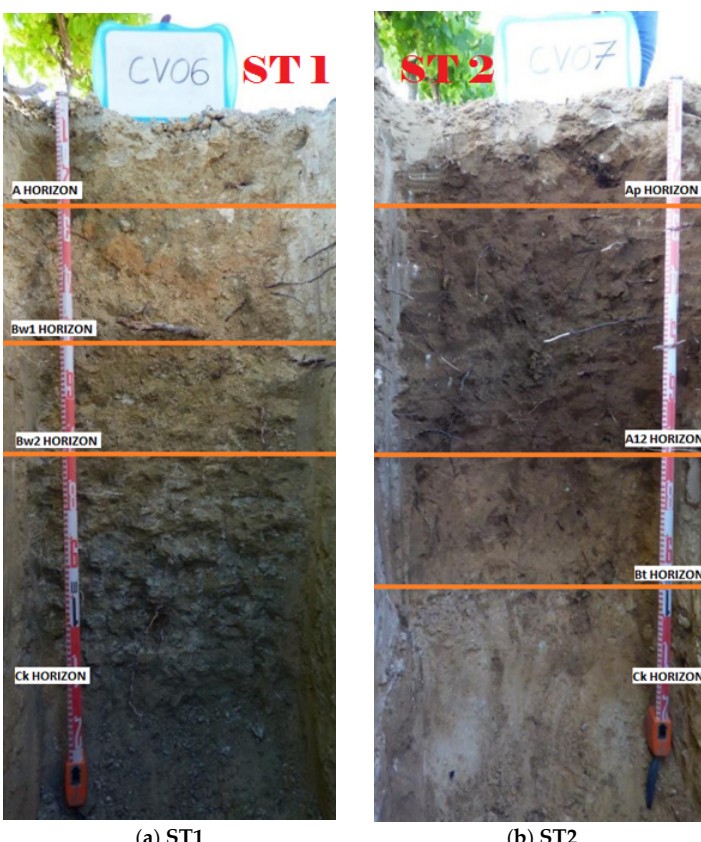

(**a**) ST1 (**b**) ST2

**Figure 6.** Horizons of two soil trenches: (**a**) Soil Trench 1 (ST1); (**b**) soil Trench 2 (ST2).

### 3.3. Soil Characteristics

The soil trenches, ST1 and ST2, both showed differences in essential parameters such as lithology, geomorphology, orientation, and slope (Table 1). The slope of both soils was similar (5.6%) and considered to be steep and slightly steep by FAO (2009). ST2 was in a concave position located 2.5 m higher than ST1. ST1 was in a convex position. The ST1 stoniness was higher than in ST2. The parental material in ST1 was detrital and carboned plasters. In ST2, it was sand and silt.

**Table 1.** Macromorphological characteristic of the horizons of two soil trenches (ST1 and ST2) located in the experimental vineyard (Toledo, Spain).

| Soil Trench | Altitude | Parent Material | Geoform | Position | Orientation | Slope (%) | Stoniness |
|---|---|---|---|---|---|---|---|
| ST1 | 569.3 a.m.s.l | Detrital and carbonated plasters | Ridged relief | Average slope (Convex − ∩) | E (79, 8°) | 5.6 | 10%, irregular limestone, 2–12 cm Ø |
| ST2 | 571.8 a.m.s.l | Sands and silts | Glacis | Low slope (Concave − ∪) | NE (41, 9°) | 5.6 | 8%, irregular limestone, 2–8 cm Ø |

Visual observation of both soil profiles (Figure 6) showed the differences in colour, horizon depth, and texture between ST1 and ST2 (Table 2). The first horizons in ST1 were a light yellowish brown, corresponding to Jaroisita soils.

**Table 2.** Physical properties of the horizons of two soil trenches (ST1 and ST2) located in the experimental vineyard (Toledo, Spain). Three repetitions of each sample.

| Soil Trench | | Texture (%) | | | Texture Class | Diagnostic Horizon | Colour |
|---|---|---|---|---|---|---|---|
| | | Clay | Silt | Sand | | | |
| ST1 | Ap (0–25 cm) | 20.7 | 30.9 | 48.4 | Loam | Ochric | 2.5Y6/3 wet and 2.5Y8/2 dry |
| | Bw1 (25–50 cm) | 13.8 | 24.4 | 61.8 | Sandy loam | Cambic | 7.5YR6/4 wet and 7.5YR7/3 dry |
| | Bw2 (50–70 cm) | - | - | - | - | Cambic | 5Y6/4 wet and 5Y7/3 dry |
| | Ck (70–130 cm) | 11.4 | 28.3 | 60.3 | Sandy loam | Calcic | 5Y5/4 wet and 5Y6/3 dry |
| ST2 | Ap (0–25 cm) | 14.0 | 21.9 | 64.1 | Sandy loam | Ochric | 10YR4/4 wet and 10YR5.5/4 dry |
| | A12 (25–70 cm) | 14.2 | 16.3 | 69.5 | Sandy loam | Ochric | 10YR3.5/3 wet and 10YR5/4 dry |
| | Bt (70–95 cm) | 17.8 | 8.8 | 73.4 | Sandy loam | Argillic | 10YR6/6 wet and 10YR7/3 dry |
| | Ck (95–125 cm) | - | - | - | Sandy loam | Calcic | 7.5YR6/8 wet and 7.5YR7/4 dry |

The first horizon in ST2 (Ap) showed a reddish matrix proper for soils with ferric properties. The A12 horizon was darker than Ap (between brown and black).

The first horizon of ST1 was an ochric diagnostic horizon (Ap). The following two horizons (Bw1 and Bw2) were cambic. ST2 was observed in two first horizon ochric (Ap and A12), and the third horizon was Cambic. The soils both presented a Ck horizon considered to be diagnostic horizon calcic.

The texture observed in the profile of both soils differed in the first horizon. Clay and silt in ST1 were 18 and 31% higher than in ST2, respectively (Table 2). Sand in ST2 was 21% higher than in ST1. The other horizons in both soils were a sandy loam texture.

The differences in texture and depth of the horizons determined the WHC (Table 3). The WHC values were 175 and 122 mm in ST1 and ST2, respectively. The WHC of ST1 was 43% higher than ST2. Moreover, the soil's average available water (AW) of the horizons were 12.3 and 7.3%, respectively. ST1 showed 74% higher gravel than ST2. The average bulk density and soil depth in both soil horizons were similar.

**Table 3.** Properties of the horizons and calculation of available water and water retention of two soil trenches (ST1 and ST2) in the experimental vineyard (Toledo, Spain). Three repetitions of each sample.

| Horizons | Depth (m) | Bulk Density (t/m3) | Available Water (% *w/w*) | Gravel (%vol) | Available Water (mm) | Water Retention(WHC) (mm) |
|---|---|---|---|---|---|---|
| ST1 | | | | | | |
| Ap | 0.25 | 1.42 | 8 | 27.8 | 28.4 | 20.5 |
| Bw1 | 0.25 | 1.45 | 11 | 38.1 | 39.9 | 24.7 |
| Bw2 | 0.15 | 1.45 | 13 | 19.5 | 37.7 | 30.4 |
| Ck | 0.50 | 1.45 | 17 | 19.5 | 123.3 | 99.2 |
| Total | 1.15 | | | | | 174.8 |
| ST2 | | | | | | |
| Ap | 0.25 | 1.45 | 9 | 11.1 | 31.3 | 27.8 |
| A12 | 0.45 | 1.45 | 8 | 9.9 | 52.2 | 47.0 |
| Bt | 0.26 | 1.45 | 6 | 3.2 | 22.6 | 21.9 |
| Ck | 0.30 | 1.45 | 6 | 3.2 | 26.1 | 25.3 |
| Total | 1.25 | | | | | 122.0 |

The soil pH was basic in both soils (>8.5) (Table 4). The saturated soil paste's electrical conductivity (ECe) in ST1 was 63% higher than in ST2. The Ck horizon in ST1 was 1.10 dS/m, classifying it as a saline horizon. The other horizons had EC values below 0.35 dS/m, which qualified them as non-saline. A saline horizon in ST1 did not affect vine production in this soil. The average organic matter (OM) contents in the horizons were 0.95 and 0.79 (g/100 g) in ST1 and ST2, respectively. The horizons' average of active limestone and cation exchange capacity (CEC) in ST1 were 42 and 63% higher than in ST2. The higher CEC values in the first horizon of ST1 were related to a higher percentage of clay, organic matter, and nitrogen. The mean concentrations of nitrogen of the horizons were 0.08 and 0.07 (g/100 g) for ST1 and ST2, respectively. The Ap horizon in ST1 had a higher N value (0.120 g/100 g).

**Table 4.** Chemical properties of horizons of two soil trenches (ST1 and ST2) located in the experimental vineyard (Toledo, Spain). Three repetitions of each sample.

| | Soil Trench | Active Limestone (g/100 g) | CE. (ext. 1:5 dS/m) | CEC (cmol(+)/kg) | pH (ext. 1:2.5 $H_2O$) | O.M. (g/100 g) | N Total (g/100 g) |
|---|---|---|---|---|---|---|---|
| ST1 | Ap (0–25 cm) | 11 | 0.21 | 13.4 | 8.6 | 2.15 | 0.120 |
| | Bw1 (25–50 cm) | 13 | 0.19 | 5.3 | 8.8 | 0.54 | 0.044 |
| | Bw2 (50–70 cm) | - | - | - | - | - | - |
| | Ck (70–130 cm) | 7 | 1.10 | 20.7 | 8.1 | 0.15 | 0.058 |
| ST2 | Ap (0–25 cm) | 5 | 0.15 | 10.3 | 8.5 | 1.45 | 0.087 |
| | A12 (25–70 cm) | 5 | 0.18 | 10.4 | 8.6 | 0.72 | 0.058 |
| | Bt (70–95 cm) | 8 | 0.22 | 4.8 | 8.6 | 0.20 | 0.037 |
| | Ck (95–125 cm) | - | - | - | - | - | - |

In summary, the differences between soils were related to water retention. The WHC of ST1 was 43% higher than ST2 due to the fine texture (clay and silt) and higher content in OM.

### 3.4. Physiological and Agronomical Parameters

The phenological evolution was different between years. Budburst occurred on 25 March 2021 and 10 April 2022, flowering on 6 May 2021 and 15 May 2022, veraison on 30 July 2021 and 21 July 2022, and harvest-ripe on 20 August 2021 and 16 August 2022.

The canopy development was different in both years (Table 5). In 2021, the ST1 vine profile was 22% higher than that of the ST2 vines. In 2022, no differences were observed.

**Table 5.** Plant vegetative growth: canopy profile, width, height, ESA, volume, gaps (%), and pruning wood of vines (PW) in two soil trenches (ST1 and ST2) located in the experimental vineyard (Toledo, Spain). Each value represents an average of six vines per soil type. SD, standard deviation.

|  | ST1 | ST2 | ST1 | ST2 |
|---|---|---|---|---|
|  | **2021** | | **2022** | |
| Profile (m) | 2.3 | 1.8 * | 1.72 | 1.82 ns |
| SD | 0.21 | 0.11 | 0.24 | 0.09 |
| Width (cm) | 61.1 | 66.1 ns | 95 | 79 ns |
| SD | 9.47 | 8.51 | 15.47 | 3.12 |
| Height (cm) | 93 | 92 ns | 94 | 86 ns |
| SD | 14.8 | 9.24 | 2.89 | 12.81 |
| ESA (m$^2$) | 2.5 | 2.4 ns | 3.1 | 2.7 ns |
| SD | 0.18 | 0.24 | 0.35 | 0.13 |
| Volume (m$^3$) | 0.69 | 0.67 ns | 0.99 | 0.75 ns |
| SD | 0.10 | 0.12 | 0.17 | 0.10 |
| Gaps (%) | 15 | 39 ** | 29.7 | 36.5 ns |
| SD | 3.11 | 5.22 | 5.11 | 5.05 |
| PW (kg/vines) | 0.18 | 0.18 ns | 0.14 | 0.19 ns |
| SD | 0.03 | 0.10 | 0.04 | 0.10 |

Levels of statistical significance (Sig.): ns, non-significant; * $p < 0.05$; ** $p < 0.01$.

In 2021, the gaps in the ST2 vines were 160% higher than in the ST1 vines. In 2022, these differences were not significant. However, gaps in the ST1 vines increased by 98% from the previous year. Gaps in the ST2 vines were similar in both years (39 and 36.5% for 2021 and 2022, respectively). The evolution of the canopy development was evaluated in 2022 (20 June 2022). No significant differences were noted between ST1 and ST2 in canopy width, height, ESA, and volume. However, for these parameters, the ST1 vines had values slightly higher than those of the ST2 vines.

No significant differences between ST1 and ST2 were observed in pruning wood (PW) weight any year. However, in 2022, the PW weight was reduced by 22 and 11% compared to 2021 in ST1 and ST2, respectively.

The SWP values at 9:00 solar time for the ST1 and ST2 vines significantly differed on 30 July 2021 and 18 August 21. In 2022, they differed only on 15 July 2022 (Table 6). On 30 July 2021, the ST1 vines had 34% higher values than the ST2 vines. Meanwhile, on 15 July 2022, the ST1 vines were 23% lower than the ST2 vines. The annual average of the ST1 vines showed a higher value in 2021 (−0.67 MPa) compared to the ST2 vines (−0.77 MPa). Thus, in 2022, the annual average of the ST1 vines was lower than that of the ST2 vines (−1.31 MPa and −1.25 MPa, respectively), but was not significantly different.

No significant differences were observed in the SWP readings obtained at 12:00 solar time in 2021. The annual averages of the ST1 and ST2 vines were −0.99 and −1.04 MPa, respectively. In 2022, there were significant differences on the first day of the irrigation season (30 June 2022), but not in the other measurements or the average. In 2022, the mean SWP was lower than in 2021 for both measurement hours.

At 9:00 hours in 2021, differences in chlorophyll between vines were observed on 18 August 2021, but not in the mean value (Table 7). However, in 2022, the ST2 vines' values were significantly higher than the values of the ST1 vines on 15 July 22 and 12 August 22.

**Table 6.** Stem water potential (MPa) measured at 9 and 12 h (solar time) on the days of the UAV flight of vines of the two soil trenches (ST1 and ST2) located in the experimental vineyard (Toledo, Spain). Each value represents an average of six vines per soil type. SD, standard deviation.

| Dates | SWP (MPa) (9 h) | | SWP (MPa) (12 h) | |
| --- | --- | --- | --- | --- |
| | ST1 | ST2 | ST1 | ST2 |
| 25 June 2021 | −0.53 | −0.57 ns | −0.88 | −0.84 ns |
| SD | 0.07 | 0.13 | 0.07 | 0.08 |
| 5 July 2021 | −0.86 | −0.88 ns | −0.84 | −0.94 ns |
| SD | 0.23 | 0.22 | 0.10 | 0.14 |
| 20 July 2021 | −0.59 | −0.64 ns | −0.99 | −0.98 ns |
| SD | 0.03 | 0.03 | 0.03 | 0.02 |
| 30 July 2021 | −0.65 | −0.87 ** | −1.01 | −1.01 ns |
| SD | 0.04 | 0.03 | 0.09 | 0.04 |
| 19 August 2021 | −0.73 | −0.90 ** | −1.3 | −1.5 ns |
| SD | 0.03 | 0.04 | 0.19 | 0.03 |
| **Average 2021** | **−0.67** | **−0.77 **** | **−0.99** | **−1.04 ns** |
| 30 June 2022 | −1.07 | −0.9 ns | −1.0 | −1.2 * |
| SD | 0.103 | 0.16 | 0.11 | 0.14 |
| 15 July 2022 | −1.17 | −0.92 * | −1.14 | −1.10 ns |
| SD | 0.16 | 0.09 | 0.14 | 0.14 |
| 5 August 2022 | −1.42 | −1.41 ns | −1.53 | −1.47 ns |
| SD | 0.13 | 0.12 | 0.10 | 0.23 |
| 12 August 2022 | −1.47 | −1.59 ns | −1.58 | −1.68 ns |
| SD | 0.09 | 0.11 | 0.18 | 0.21 |
| **Average 2022** | **−1.31** | **−1.25 ns** | **−1.43** | **−1.47 ns** |

Levels of statistical significance (Sig.): ns, non-significant; * $p < 0.05$; ** $p < 0.01$.

**Table 7.** Leaf chlorophyll content (micromol/m$^2$ foliar surface), measured at 9 and 12 h (solar time) on the days of the UAV flight, of vines of the two soil trenches (ST1 and ST2) located in the experimental vineyard (Toledo, Spain). Each value represents an average of six vines per soil type. SD: Standard Deviation.

| Dates | Chl (Micromol/m$^2$) (9 h) | | Chl (Micromol/m$^2$) (12 h) | |
| --- | --- | --- | --- | --- |
| | ST1 | ST2 | ST1 | ST2 |
| 25 June 2021 | 17.1 | 17.6 ns | 14.90 ns | 15.40 ns |
| SD | 3.43 | 2.56 | 2.64 | 4.07 |
| 5 July 2021 | 16.1 | 19.9 ns | 17.53 | 17.25 ns |
| SD | 4.29 | 1.87 | 2.15 | 6.62 |
| 20 July 2021 | 17.6 | 20.0 ns | 19.73 | 21.38 ns |
| SD | 1.52 | 2.11 | 2.80 | 2.69 |
| 30 July 2021 | 17.9 | 19.3 ns | 19.88 | 20.43 ns |
| SD | 0.67 | 4.37 | 2.68 | 3.99 |
| 19 August 2021 | 16.2 | 20.5 * | 21.00 | 20.80 ns |
| SD | 1.96 | 2.75 | 2.89 | 3.02 |
| **Average 2021** | **18.63** | **19.08 ns** | **16.98** | **19.45 ns** |
| 30 June 2022 | 13.29 | 15.64 ns | 13.50 | 16.53 * |
| SD | 1.71 | 2.75 | 1.12 | 1.39 |

**Table 7.** *Cont.*

| Dates | Chl (Micromol/m$^2$) (9 h) | | Chl (Micromol/m$^2$) (12 h) | |
|---|---|---|---|---|
| | ST1 | ST2 | ST1 | ST2 |
| 15 July 2022 | 14.71 | 17.63 ** | 14.20 | 17.75 ** |
| SD | 1.02 | 1.32 | 1.26 | 1.20 |
| 5 August 2022 | 15.96 | 18.02 ns | 16.47 | 18.46 ns |
| SD | 1.78 | 1.95 | 1.56 | 1.12 |
| 12 August 2022 | 13.87 | 17.18 * | 14.95 | 17.11 ns |
| SD | 1.16 | 3.05 | 1.51 | 2.65 |
| **Average 2022** | **14.76** | **17.30 **** | **15.03** | **17.83 **** |

Levels of statistical significance (Sig.): ns, non-significant; * $p < 0.05$; ** $p < 0.01$.

The chlorophyll values at 12 h were increased in all vines compared with the 9 h measurements. In 2021, no significant differences were observed between ST1 and ST2. In 2022, SWP readings in the ST2 vines were 19% higher than in the ST1 vines.

At both hours of measurement, chlorophyll was lower in 2022 compared to 2021.

In 2021, the ST1 vines produced a significantly higher number of bunches (72%), and yield was three times higher than in ST2 (Table 8), but not in 2022. Concerning quality, significant differences were only observed in total acidity in 2021. Vines in ST1 presented 17% higher acidity than those in ST2.

**Table 8.** Yield parameters and maturity parameters. Bunches, number of bunches per vine. Yield (kg/vine). SC, soluble solids content (°Brix); pH; TA, total acidity (g of tartaric·L-1) of vines of the two soil trenches (ST1 and ST2) located in the experimental vineyard (Toledo, Spain). Each value represents an average of six vines per soil type. SD, standard deviation.

| | ST1 | ST2 | ST1 | ST2 |
|---|---|---|---|---|
| | **2021** | | **2022** | |
| Bunches (#) | 71 | 20 ** | 63 | 56 ns |
| SD | 12.3 | 19.6 | 18.7 | 19.7 |
| Yield (kg/vines) | 3.56 | 1.22 * | 1.54 | 2.10 ns |
| SD | 0.75 | 1.04 | 0.68 | 0.79 |
| SST (°Brix) | 26.7 | 29.3 ns | 26.6 | 27.0 ns |
| SD | 0.76 | 2.04 | 0.3 | 1.6 |
| pH | 3.37 | 3.48 ns | 3.43 | 3.41 ns |
| SD | 0.05 | 0.08 | 0.38 | 0.09 |
| TA | 6.34 | 5.26 ** | 5.24 | 5.33 ns |
| SD | 0.44 | 0.22 | 0.35 | 0.18 |

Levels of statistical significance (Sig.): ns, non-significant; * $p < 0.05$; ** $p < 0.01$.

### 3.5. Soil Thermal and Spectral Characteristics

Soil temperature measured with the thermal camera significantly differed between soils (Table 9). The average temperature difference between the two soil trenches for both years was 1.8 °C. ST1 presented lower temperatures than ST2 in both seasons. The most significant differences were observed on 20 July 2021 between soils (ST1 = 53 °C and ST2 = 56 °C). The maximum value was observed on 20 July 2021, with 3 °C difference. The mean temperature of ST2 was 2.5 °C higher than that of ST1.

**Table 9.** The soil surface temperature of nearby experimental vines (ST1 and ST2) in the vineyard (Toledo, Spain). Each value represents an average of 300 pixels for each soil type. SD, standard deviation.

| Soil | Temperature | | | |
|---|---|---|---|---|
| | **17 May 2021** | **20 July 2021** | **19 May 2022** | **15 July 2022** |
| ST1 | 43.4 | 52.9 | 51.1 | 59.3 |
| SD | 0.76 | 2.04 | 1.04 | 1.27 |
| ST2 | 45.2 | 55.9 | 51.4 | 61.6 |
| SD | 0.75 | 1.43 | 1.17 | 1.43 |
| Sig | ** | ** | * | ** |

Levels of statistical significance (Sig.): ns, non-significant; * $p < 0.05$; ** $p < 0.01$.

The multispectral response differed between soils (Figure 7). The distribution of the pixel histogram in the red edge, NIR, and green bands showed a clear separation between soils in both years. In 2021, no significant differences were observed in the red band, while in 2022, differences did appear.

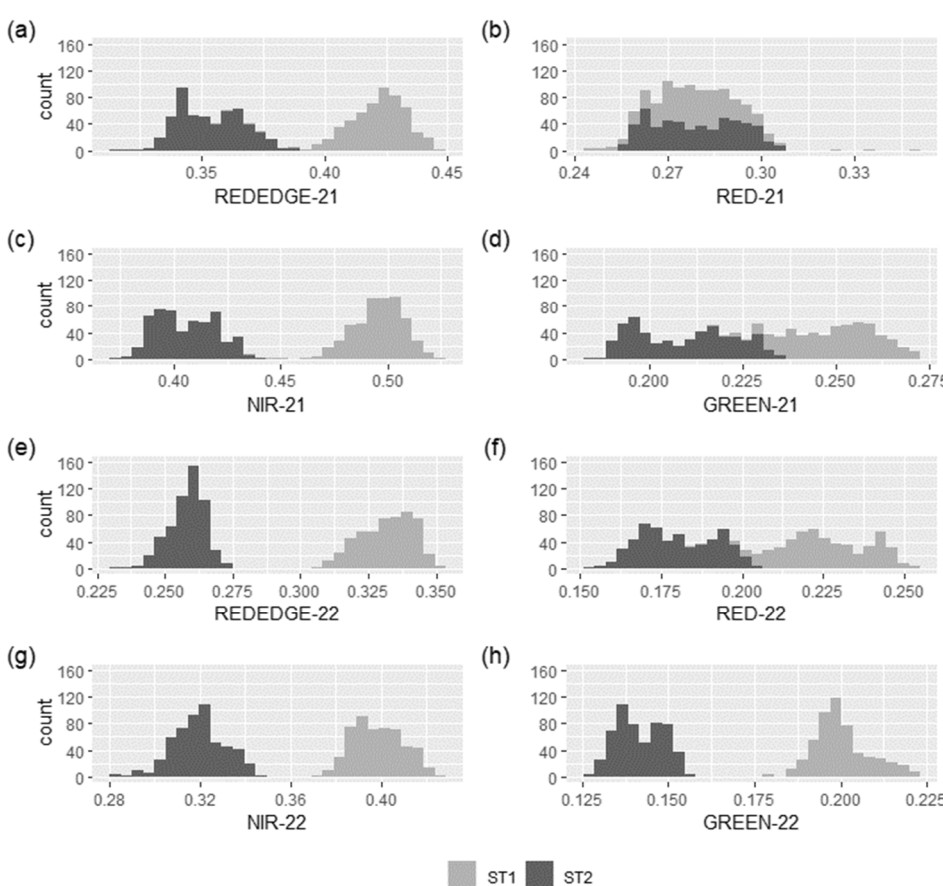

**Figure 7.** Histogram of each multispectral band of the soil surface of nearby experimental vines (ST1 and ST2) in the vineyard (Toledo, Spain) based on 500 pixels for each site. The first two rows correspond to 2021 (−21), and the rest correspond to 2022 (−22). The bands are red edge (**a**,**e**); red (**b**,**f**); NIR (**c**,**g**); and green (**d**,**h**).

Band reflectance values for the ST1 soil were significantly higher compared to ST2. The NIR, green, and red edge reflectance of ST1 were 18, 16, and 16% higher than those of the ST2 vines, respectively. The maximum difference values were observed in the red edge band (ST1 = 0.42 and ST2 = 0.35) in 2021.

In 2022, the red, NIR, green, and red edge bands of ST1 were 20, 20, 29, and 22% higher than those of the ST2 vines. The most significant differences between vines were observed on the green band (ST1 = 0.20 and ST2 = 0.14). The green and red edge bands showed the highest differences between soils in both years.

The green band showed these differences with some overlap between soil pixels, but the soils were still significantly different in 2021. The NDVI, RVI, and GNDVI values showed significant differences between soils in 2021 (Figure 8). In ST1, they were 34, 18, and 59% higher than in ST2. In 2022, only the GNDVI value showed significant differences. ST1 was 85% higher than ST2. No significant differences were observed in the NDVI and RVI values.

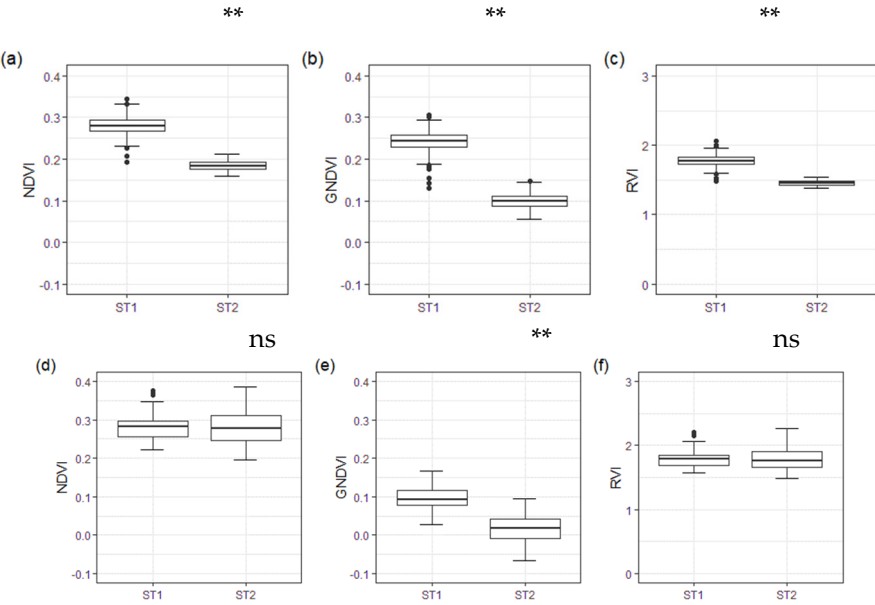

**Figure 8.** Vegetation indices of two soil trenches (ST1 and ST2) located near the experimental vines (Toledo, Spain). The first row corresponds to 2021, and the second row corresponds to 2022. The indices are NDVI (**a,d**); RVI (**b,e**); and GNDVI (**c,f**). Levels of statistical significance (Sig.): ns, non-significant; * $p < 0.05$; ** $p < 0.01$. Statistics are based on 500 pixels for each soil trench (ST1 and ST2).

In summary, the differences in soil spectra images did not depend on the year. The green, red edge, and NIR bands differentiated the two soil types. The largest VI differences between soils were observed in the GNDVI.

### 3.6. Vegetation Spectral Characteristics

The VI values obtained from the multispectral images of vine canopies differed within the vines in ST1 and ST2 soils (Table 10). In 2021, all VI values of the ST1 vines were significantly higher than those of the ST2 vines. The maximum values of four of the VIs calculated were found on 20 July 2021 in the ST1 vines (NDVI = 0.83, RVI = 10.48, OSAVI = 0.74, and GNDVI = 0.76).

The RVI, NDVI, and OSAVI values of the ST1 vines were 55, 10, and 11% higher than those in the ST2 vines. The lowest values of the VIs occurred in the S2 vines (NDVI = 0.73, RVI = 6.5, and OSAVI = 0.64) at harvest, and the differences between vines were reduced. At this measure, the VI values of the ST1 vines were 5, 17, and 3% higher than those of the ST2 vines for the NDVI, RVI, and OSAVI, respectively. The mean 2021 values of the NDVI, RVI, and NDRE were significantly different between the ST1 and ST2 vines.

**Table 10.** VIs at 12 h (solar time) in the days of the UAV flight of vines of the two soil trenches (ST1 and ST2) located in the experimental vineyard (Toledo, Spain). Each value represents an average of 4–5 pixels/vines for six vines per soil type. SD, standard deviation.

| Dates | Vines | NDVI | RVI | OSAVI | GNDVI | NDRE |
|---|---|---|---|---|---|---|
| 25 July 2021 | ST1 | 0.84 | 11.33 | 0.75 | 0.74 | 0.20 |
| | ST2 | 0.75 | 7.07 | 0.65 | 0.69 | 0.22 |
| | sig. | ** | ** | ** | ** | ** |
| SD | ST1 | 0.011 | 0.793 | 0.013 | 0.009 | 0.013 |
| | ST2 | 0.046 | 1.208 | 0.051 | 0.029 | 0.007 |
| 5 July 2021 | ST1 | 0.79 | 8.56 | 0.69 | 0.71 | 0.24 |
| | ST2 | 0.79 | 8.55 | 0.70 | 0.73 | 0.24 |
| | sig. | ns | ns | ns | ** | ns |
| SD | ST1 | 0.004 | 0.200 | 0.008 | 0.005 | 0.015 |
| | ST2 | 0.019 | 0.782 | 0.021 | 0.011 | 0.008 |
| 20 July 2021 | ST1 | 0.83 | 10.48 | 0.74 | 0.76 | 0.24 |
| | ST2 | 0.77 | 7.72 | 0.69 | 0.74 | 0.26 |
| | sig. | ** | ** | ** | ** | ** |
| SD | ST1 | 0.007 | 0.465 | 0.010 | 0.008 | 0.011 |
| | ST2 | 0.030 | 1.005 | 0.034 | 0.019 | 0.013 |
| 30 July 2021 | ST1 | 0.73 | 6.48 | 0.64 | 0.73 | 0.21 |
| | ST2 | 0.69 | 5.46 | 0.60 | 0.68 | 0.21 |
| | sig. | ** | ** | ** | ** | ns |
| SD | ST1 | 0.007 | 0.200 | 0.014 | 0.010 | 0.010 |
| | ST2 | 0.031 | 0.591 | 0.032 | 0.035 | 0.018 |
| 19 August 2021 | ST1 | 0.77 | 7.85 | 0.67 | 0.71 | 0.20 |
| | ST2 | 0.73 | 6.64 | 0.65 | 0.71 | 0.22 |
| | sig. | ** | ** | ** | ns | ** |
| SD | ST1 | 0.016 | 0.598 | 0.021 | 0.009 | 0.017 |
| | ST2 | 0.037 | 1.001 | 0.032 | 0.028 | 0.020 |
| **Average 2021** | **ST1** | **0.79** | **8.87** | **0.69** | **0.73** | **0.22** |
| | **ST2** | **0.74** | **6.95** | **0.65** | **0.71** | **0.23** |
| | **sig** | **** | **** | **ns** | **ns** | **\*** |
| 30 June 2022 | ST1 | 0.77 | 7.57 | 0.58 | 0.65 | 0.19 |
| | ST2 | 0.77 | 7.87 | 0.58 | 0.65 | 0.21 |
| | sig. | ns | ns | ns | ns | ** |
| SD | ST1 | 0.018 | 0.58 | 0.02 | 0.01 | 0.01 |
| | ST2 | 0.038 | 1.15 | 0.03 | 0.03 | 0.01 |
| 15 July 2022 | ST1 | 0.74 | 6.70 | 0.57 | 0.65 | 0.24 |
| | ST2 | 0.73 | 6.55 | 0.56 | 0.65 | 0.20 |
| | sig. | ns | ns | ns | ns | ** |
| SD | ST1 | 0.02 | 0.45 | 0.01 | 0.01 | 0.02 |
| | ST2 | 0.03 | 0.78 | 0.03 | 0.02 | 0.02 |
| 5 August 2022 | ST1 | 0.73 | 6.32 | 0.55 | 0.59 | 0.24 |
| | ST2 | 0.73 | 6.38 | 0.57 | 0.62 | 0.28 |
| | sig. | ns | ns | ** | ** | ** |
| SD | ST1 | 0.02 | 0.56 | 0.02 | 0.02 | 0.01 |
| | ST2 | 0.02 | 0.40 | 0.02 | 0.01 | 0.01 |
| 12 August 2022 | ST1 | 0.66 | 4.86 | 0.51 | 0.58 | 0.24 |
| | ST2 | 0.64 | 4.56 | 0.49 | 0.58 | 0.17 |
| | sig. | ** | ** | ** | ns | ** |
| SD | ST1 | 0.02 | 0.33 | 0.02 | 0.02 | 0.02 |
| | ST2 | 0.02 | 0.25 | 0.02 | 0.01 | 0.01 |
| **Average 2022** | **ST1** | **0.72** | **6.31** | **0.55** | **0.61** | **0.23** |
| | **ST2** | **0.72** | **6.33** | **0.55** | **0.63** | **0.22** |
| | **sig.** | **ns** | **ns** | **ns** | **ns** | **\*** |

Levels of statistical significance (Sig.): ns, non-significant; * $p < 0.05$; ** $p < 0.01$.

The VI values distribution drastically changed in 2022. The mean value of the indices between vines was similar but lower compared to 2021. At the beginning of the 2022 season (30 June), the NDVI, RVI, OSAVI, GNDVI, and NDRE values in the ST1 vines decreased by 9, 33, 23, 13, and 5%, respectively, compared with those in 2021. Near harvest (12 August), in 2022, the NDVI, RVI, OSAVI, and GNDVI values of the ST1 vines decreased by 15, 38, 24, and 19% compared to the previous season. Conversely, the NDRE of the ST1 vines increased by 20% in 2022.

Differences between the evaluated vine VIs depended on the season. The RVI showed the highest differences between vines on different soils in the 2021 season. In 2021, the RVI value of the ST1 vines was 22% higher than that of the ST2 vines. In 2022, the NDRE index was the only index that showed differences between the vines.

## 4. Discussion

Soil zoning of vineyards can be undertaken on different scales based on the objective. Soil often has a tremendous spatial variability; a small-scale mapping approach will encounter difficulties considering this soil variability [14].

In our work, photo interpretation allowed us to establish a preliminary classification of seven different soils in the study zone (Figure 3a). Information extracted from soil sampling in the field helped us to adjust the study zone to consider just two types of soils: ST1 and ST2 (Figure 3b). The soils were different in some essential characteristics, such as texture and depth. Differences in these parameters between horizons significantly affected vine development, especially on the WHC [3].

In non-irrigated plots, soil characteristics impacted grapevine water status, yield, and fruit composition [37,38]. Under irrigation conditions, the effect of soil properties on vine water status should not be strong, because the water supply to the plant is guaranteed more by irrigation than by the water reservoir in the soil [4]. The study area was characterised by low rainfall and high evaporation, and deficit irrigation was applied. The differences between years in water available provoked a temporal heterogeneity. Rain + irrigation was reduced by 49% in 2022 compared to 2021 (Figure 2), indicating that water-holding capacity in soil is decisive for the development of plants. Our study observed that ST1 was 43% higher in terms of the WHC than ST2 (Table 3).

The average WHC values in ST1 and ST2 were 175 and 122 mm, respectively. These value were considered to be a medium range. Ref. [39] established that this parameter on vineyards was highly variable, covering a range from 50 mm in very shallow soils with a sandy texture and with a high percentage of coarse elements, to over 350 mm in silty soils that allow deep rooting. The water retention capacity in the soil is affected by soil properties, such as gravel content, leading to a higher incidence of water deficits in vine growth [40]. However, we observed the opposite in our study. The average gravel content in the ST1 horizons was 74% higher than that in ST2. This suggests that gravel was not relevant to estimating the WHC. The main properties that differed between soils and influenced WHC were the higher OM content and percentage of finer texture (clay and silt) in ST1 compared to ST2. According to [3], soil water is stored in porosity. The size of the pores has an essential influence on the availability to plants. Pore size varies with soil texture; hence, soil texture significantly impacts the soil WHC. Ref. [40] compared the quantitative measures of soil water retention capacity for two soils on opposing slopes and obtained similar results. Their study evaluated north aspect slopes compared with south aspect slopes. Soil porosity, soil OM, and silt content were all greater on the north aspect.

Regarding ECe, the ST1 vines had a layer (Ck) that was considered saline (1.10 dS/m). Under irrigation deficit, this horizon may affect the yield of the vines in ST1. Ref. [4] observed that areas with maximum ECe values of 2.0 dS/m could have limited root water uptake.

However, the influence of soil properties on grapevine water statuses also differs depending on the climatic conditions and irrigation [38]. Climate affects the vine water status through rainfall and reference crop evapotranspiration (ETo) [39]. Climatic conditions

during the two years of the study were highly evaporative, especially in 2022, which was particularly dry due to less precipitation and irrigation (the irrigation amount was reduced by 39%).

These yearly differences in climatic conditions could explain differences in plant development. Vine vigour is often reported to affect fruit yield and quality considerably [1]. Thus, measures of vine canopy can be used to estimate differences in fruit yield and quality. Poor vegetative development directly influences yield and maturity parameters. In 2021, the ST1 vines had higher performances in terms of the number of bunches and TA compared to the ST2 vines. The yield in the ST1 vines was three times higher than in the ST2 vines (Table 8). However, in 2022, this changed. The zones both saw this parameter reduced, excluding the brix and pH in the ST1 vines, which were maintained. The ST2 vines showed a slightly reduced Brix degree (from 29.4 to 26.6), and the pH remained stable.

SWP represents the whole vine water status during the day and is a particularly useful tool for irrigation management. It accurately represents vine water status, even if the soil water content is heterogeneous [39]. In 2021, the studied vines showed average SWP readings of $-0.99$ and $-1.04$ MPa for the midday measures in the ST1 and ST2 vines, respectively, which were categorised as moderate to weak water deficits [39]. However, in 2022, the SWP readings were $-1.43$ and $-1.47$ MPa in the ST1 and ST2 vines, respectively, categorised as a moderate to severe water deficit due to the increased evaporative conditions and lower irrigation. Although the water stress conditions were similar for the vines of the two soils in 2022, the effect was visible in their vegetative performance. This suggested the superior adaptation of the ST2 vines to stress conditions. Ref. [41] observed that vines exposed to early water deficit stress developed smaller xylem vessels than vines developed under unlimited water uptake conditions. In agreement with [39], irreversible embolisms might occur in vigorous vines with large xylem vessels that are suddenly exposed to excessive water deficit SWP readings of $-1.2$ MPa. Low vigour vines progressively exposed to water deficits might show SWP levels of $-1.6$ MPa. Limited vine water uptake could reduce shoot growth and yield. Thus, the ST2 vines with lower vigour in 2021 resisted the deficit conditions in 2022, maintaining yield and vegetative development. However, the ST1 vines showed a reduced profile, volume, and surface leaf area, and increased canopy gaps in 2022 compared to in 2021, which meant less vegetative development. Water deficit stress during the season causes the stomata to close for part of the day [42]. This restricts photosynthesis. Hence, dry matter production is reduced, which explains the pruning weight and yield results in 2022 compared with in 2021.

The chlorophyll content of the studied vines (Table 7) showed some interesting features. The ST1 vines had lower values than the ST2 vines in both years and in both hours of measurement. The soils' chemical properties could explain this. The studied soils have a significant percentage of active limestone (Table 4), especially ST1. However, this did not influence the yield and quality of the vines in 2021 (with less hydric restriction). Instead, high calcium improves the soil structure [16], improving root penetration, speeding up soil warming in the spring and improving internal drainage. Active lime also reduces soil organic matter in turnover, limiting mineral nitrogen availability. Ref. [43] suggested this could affect chlorophyll production. Thus, the ST1 vines had lower chlorophyll values than the ST2 vines.

According to [1], vigour can indicate the effect of environmental factors and management on vines. Vigour can be easily mapped using remote sensing as a zoning tool. The NDVI, RVI, OSAVI, GNDVI, and NDRE indicated differences between the ST1 and ST2 vines in 2021. In 2021, the annual average values of the NDVI, RVI, and NDRE were significantly different between the ST1 and ST2 vines (Table 10). ST1 presented higher values for each VI and date, corresponding to higher vegetation development. Excluding NDRE, the average of the ST2 vines in 2021 was slightly higher than the ST1 vines. The VI values in 2022 did not show significant differences between vines. However, because there was no clear trend among the NDRE values for the ST1 and ST2 vines during the two years of the study, no conclusions can be drawn.

The vine RVI values showed unusual behaviour. Unlike the other indices, a trend was observed in the RVI values in 2021. When the season began, the values for both vines were higher, but these values decreased as the season progressed. This index showed the lowest value on the last day (18 August), close to harvest. Overall, the RVI showed higher differences among the VI values evaluated in terms of the annual average. In the ST1 vines, the RVI was 22% higher than in the ST2 vines. In 2022, although the difference was less evident, the index showed a similar trend. At the beginning of the season, the values of the RVI were higher (but lower compared to the beginning of 2021). These values remained stable during the middle dates (15 July and 05 August) and later decreased, as in 2021. Ref. [20] observed the variability of the indices of vines through a flight campaign in their study zone. In this study, an overall NDVI decline was noted from the third flight of the campaign onward. This was related to the grapevines' vegetative cycle. The maturity stage of the leaf better explained the reduction in the VI values than the decrease in vigour.

The two soils showed differentiated behaviour in their band reflection values at the multispectral level (Figure 7). Values of the NIR, red edge, and green bands best differentiated the soils in both years. The red band did not present differences between the two soils, especially in 2021. The band's behaviour for the soils suggested that components in their structure influence a greater or lesser radiation absorption in those bands. The combination of fundamental features of the soil has overtones that can cause spectral signatures in the visible and NIR regions, making the visible and NIR regions potentially helpful in determining many soil components [13]. A correlation between physical properties, such as the percentage of clay and sand, with the green visible band was observed by [15]. These findings validated the observed differences in the VI values, where the GNDVI exhibited significant variations between soils compared to the NDVI or RVI, primarily due to the influence of the green band (Figure 8). To comprehend this, one must consider the variations in soil texture between the studied soils. Specifically, ST1 has a higher clay content in its initial horizons than ST2 (Table 2).

Thermal sensor data were obtained to complement the soil information (Table 9). Soil temperature depends on the energy balance, which is related to soil colour and albedo (proportion of sunlight reflected on the soil), slope steepness, and slope direction [3]. It can be measured, but because it is spatially and temporally variable, it is challenging to compute as a relevant indicator. Warm and cool soils can be identified by expertise, as warm soils tend to be coarse textured and high in coarse elements [3,44]. Thus, ST1 was colder than ST2, with a difference of 2.5 °C in summer. The maximum value was observed on 20 July 2021, with 3 °C of difference. On the one hand, this suggested that more than the gravel elements present in ST1, the fine texture, or colour could have influenced the lower temperature. On the other hand, the colour of the soil may have had an influence, as ST1 was a lighter soil, reflected more energy, and therefore warmed up less than the darker ST2 soil.

These results indicated the effectiveness of using UAVs for identifying different management zones. Implementing high-resolution imagery focused on grapevine vegetation precisely depicted the vineyard's variability.

Several authors have shown the usefulness of IVs such as the NDVI to delineate areas with heterogeneous vigour within vineyards, which correspond with different plant performances [1,44–47].

However, yield maps or management zone maps based on the vines' vegetative growth should be considered under non-stress conditions; otherwise, the variability of the vines may go unnoticed.

Taking into consideration soil information at the sampling level and using UAVs allowed more precise zoning maps to be made.

## 5. Conclusions

Spatial and temporal heterogeneity is a reality that vine growers face year after year. The soil influences spatial heterogeneity, while climate influences temporal heterogeneity.

For optimal vineyard management, proper soil zoning is necessary. Knowledge of soil heterogeneity will enable the different management of plots. This involves adapting cultural practices to specific characteristics, such as the irrigation system, spacing between drippers, flow rate, soil management, pruning, and harvesting date. As an alternative to traditional methods of zoning map generation, information obtained from remote sensors allows for quick and easy observation of the spatial and temporal heterogeneity.

In this experiment, temporal heterogeneity was found to be provoked by the differences in water availability between years. Rain + irrigation was reduced by 49% in 2022 compared to that in 2021. There was spatial heterogeneity between the evaluated soils. The WHC of ST1 was 43% higher than that of ST2 due to the fine texture (clay and silt) and higher OM. In 2021, vines in soil with higher WHC (ST1) had yields that were three times higher than those of the ST2 vines.

High-resolution images from the multispectral camera onboard the UAV allowed us to calculate different vine VI values. In 2021, significant differences between vine VI values in different soils (NDVI, RVI, and NDRE) were detected. The RVI showed the most significant differences of the VIs evaluated, where, in terms of the annual average, ST1 vines were 22% higher than ST2 vines. Meanwhile, the 2022 NDRE index was the only VI that was statistically different in the drier year.

Spectral images of the soils showed differences in both years due to their physical properties (colour and texture). Specifically, the green, red edge, and NIR bands successfully differentiated the two soil types. The largest VI differences between soils were observed in the GNDVI. The thermal images showed significant differences between soils. The temperature of ST2 was 2.5 °C higher than that of ST1.

Regardless of the climatic conditions, soil spectral and soil thermal characteristics are a reliable source of information, making them a more robust element for zoning than the vine vegetation itself. These results confirmed that UAVs are a valuable tool for assessing spatial and temporal heterogeneity and monitoring vineyards at minimal operational expense. Regarding zoning applicability, soil spectral and thermal information is essential, as it is influenced much less between years than vegetative information.

**Author Contributions:** Conceptualisation, L.K.A.P., A.M.T. and M.G.d.C.; data curation, L.K.A.P., A.M.T. and M.G.d.C.; formal analysis, A.M.T.; investigation, L.K.A.P., A.M.T. and M.G.d.C.; methodology, L.K.A.P., A.M.T., R.H.P., J.C. (Jesus Cano), J.C. (Joaquin Cámara) and M.G.d.C.; writing—original draft, L.K.A.P., J.N. and M.G.d.C.; writing—review and editing, L.K.A.P., A.M.T. and M.G.d.C. All authors have read and agreed to the published version of the manuscript.

**Funding:** Financial support was provided by *Comunidad de Madrid* through calls for grants for the completion of Doctorado Industrial IND2020/AMB-17341 and was greatly appreciated.

**Data Availability Statement:** Not applicable.

**Acknowledgments:** The authors thank *Bodegas y Viñedos Casa del Valle* for allowing us to work in their vineyards, and the company UTW for supplying technical support and drone images.

**Conflicts of Interest:** The authors declare no conflict of interest.

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
