# Peer review of "Multispectral and Thermal Sensors Onboard UAVs for Heterogeneity in Merlot Vineyard Detection: Contribution to Zoning Maps"

_remotesensing, doi:10.3390/rs15164024_

Round 1
Reviewer 1 Report
I appreciate the opportunity to provide a peer review of the manuscript entitled, “Assessment of soil influence on a vineyard with multispectral and thermal sensors onboard UAVs: A case study of the Merlot variety in Yepes-Toledo.” currently in submission to Remote Sensing. Overall, the manuscript provides an overview of a study evaluating the influence of soil types on vine development in a vineyard using UAVs and multispectral data. Soil properties studies are essential for farmers because they assist with decision-making relative to more crop yield and productivity. Therefore, this is important work, of high potential relevance and could be an appropriate fit for the journal’s readership. However, the manuscript has many grammatical, flow, and content problems, making it very challenging to follow the work. Given the potential value of this work, I strongly recommend that the authors develop the manuscript before resubmission. I supply additional comments as follows:
1. The abstract is somewhat nebulous and needs to be improved in terms of the focus of the study and the precision of results and findings. i.e., the authors should ensure that the abstract is clear, brief, concise, and grammatically correct. For example, the abstract briefly mentions the use of multispectral data, vegetation indices, stem water potential, chlorophyll content, and yield parameters. More specific details about the measurements and techniques should be employed. The authors should present salient results in the abstract to attract the reader.
2. Objectives should be restated clearly at the end of the introduction. Clearly articulate what the study aims to investigate or analyze regarding the influence of soil types on vine development.
3. Many methods in the manuscript lack specificity, such as the number of vines in the experimental plots and the amount of data collected for each measurement. Regarding Tables 2-4, How many plots (soil points) were used? I suggest adding a flowchart to enhance readability and provide a clear overview of the experimental process.
4. The study briefly mentions that the results show the effect of soils on vine development under water-limited conditions and that UAVs help detect changes in vines and identify different soil types. However, it does not provide any specific insights or implications resulting from the study. Consider discussing the practical implications of the findings and how they contribute to the understanding of vineyard management and water optimization strategies.
5. There are many grammatical and flow problems in the text that I will not list here but that the author should be sure to correct. I strongly recommend that the author retain an English-speaking editor. In its current form, the text is rife with convolutions and extremely confusing statements that make it very challenging to understand. It is challenging to follow the investigation because the presentation of the work is not yet well written.
6. The standard deviation needs to be included in Tables 5-10. Lines 559-561???? This is a very fragmented manuscript. This sentence is suitable for discussion section.
7. The authors should make sure that the conclusions include 1) a problem statement, 2) an approach to solving the problem with great brevity, 3) the most salient results, and 4) implications (in brief).
I applaud the authors for undertaking this ambitious work and the venture of extracting something for a prestigious peer review journal such as Remote Sensing. I recommend the authors continue to hone the research/literature review, analysis, and presentation/writing and resubmit when ready.
Extensive editing of English language required
Author Response
I appreciate the opportunity to provide a peer review of the manuscript entitled, “Assessment of soil influence on a vineyard with multispectral and thermal sensors onboard UAVs: A case study of the Merlot variety in Yepes-Toledo.” currently in submission to Remote Sensing. Overall, the manuscript provides an overview of a study evaluating the influence of soil types on vine development in a vineyard using UAVs and multispectral data. Soil properties studies are essential for farmers because they assist with decision-making relative to more crop yield and productivity. Therefore, this is important work, of high potential relevance and could be an appropriate fit for the journal’s readership. However, the manuscript has many grammatical, flow, and content problems, making it very challenging to follow the work. Given the potential value of this work, I strongly recommend that the authors develop the manuscript before resubmission. I supply additional comments as follows:
Thank you very much for your suggestions. We have corrected in the manuscript the issues stated by the reviewer. All the changes are in yellow in the new version of the manuscript.
- The abstract is somewhat nebulous and needs to be improved in terms of the focus of the study and the precision of results and findings. i.e., the authors should ensure that the abstract is clear, brief, concise, and grammatically correct. For example, the abstract briefly mentions the use of multispectral data, vegetation indices, stem water potential, chlorophyll content, and yield parameters. More specific details about the measurements and techniques should be employed. The authors should present salient results in the abstract to attract the reader.
The focus of the study has been improved and the main results have been highlighted in the abstract.
- Objectives should be restated clearly at the end of the introduction. Clearly articulate what the study aims to investigate or analyze regarding the influence of soil types on vine development.
Done
- Many methods in the manuscript lack specificity, such as the number of vines in the experimental plots and the amount of data collected for each measurement.
Done
Regarding Tables 2-4, How many plots (soil points) were used? I suggest adding a flowchart to enhance readability and provide a clear overview of the experimental process.
Done
- The study briefly mentions that the results show the effect of soils on vine development under water-limited conditions and that UAVs help detect changes in vines and identify different soil types. However, it does not provide any specific insights or implications resulting from the study. Consider discussing the practical implications of the findings and how they contribute to the understanding of vineyard management and water optimization strategies.
The practical implications of the study are summarized in a paragraph in the conclusions.
- There are many grammatical and flow problems in the text that I will not list here but that the author should be sure to correct. I strongly recommend that the author retain an English-speaking editor. In its current form, the text is rife with convolutions and extremely confusing statements that make it very challenging to understand. It is challenging to follow the investigation because the presentation of the work is not yet well written.
The text has been revised, and a flowchart has been incorporated to enhance the clarity of the content and facilitate the understanding of the process.
- The standard deviation needs to be included in Tables 5-10.
Lines 559-561???? This is a very fragmented manuscript. This sentence is suitable for discussion section
These lines were removed.
- The authors should make sure that the conclusions include 1) a problem statement, 2) an approach to solving the problem with great brevity, 3) the most salient results, and 4) implications (in brief).
Done
Comments on the Quality of English Language
Extensive editing of English language required
Thank you for your valuable feedback. We sincerely appreciate your input and hope the updated version meets your expectations.
Reviewer 2 Report
Following are my observations in this article which are required in order to attain the high readability goal.
1. Title needs to be revised.
2. In abstract must write something about what you have achieved if compare with other ones.
3. Introduction must be split into subparts i.e., (Motivation, Related Work, Contributions and Organization).
4. Need to add Heading of problem statement and the proposed solution.
5. Need to add another heading State of the ART work.
6. In Figure 1 must mention Year in x axis
7. Check equation (3)
8. Must make a comparative table of figure 5
Following are my observations in this article which are required in order to attain the high readability goal.
1. Title needs to be revised.
2. In abstract must write something about what you have achieved if compare with other ones.
3. Introduction must be split into subparts i.e., (Motivation, Related Work, Contributions and Organization).
4. Need to add Heading of problem statement and the proposed solution.
5. Need to add another heading State of the ART work.
6. In Figure 1 must mention Year in x axis
7. Check equation (3)
8. Must make a comparative table of figure 5
Author Response
Following are my observations in this article which are required in order to attain the high readability goal.
Thank you very much for your suggestions. We have corrected in the manuscript the issues stated by the reviewer. All the changes are in yellow in the new version of the manuscript.
- Title needs to be revised.
change the title to:
“Multispectral and thermal sensors onboard UAVs for heterogeneity in Merlot vineyard detection. Contribution to zoning maps”
- In abstract must write something about what you have achieved if compare with other ones.
Done
- Introduction must be split into subparts i.e., (Motivation, Related Work, Contributions and Organization).
Done
- Need to add Heading of problem statement and the proposed solution.
Done
- Need to add another heading State of the ART work.
Done
- In Figure 1 must mention Year in x axis
Done
- Check equation (3)
Done
- Must make a comparative table of figure 5.
The figure 5 has been relocated next to the descriptive tables for a better understanding of the soil.
Thank you for your valuable feedback. We sincerely appreciate your input and hope the updated version meets your expectations.
Reviewer 3 Report
The authors conducted a study of soil characteristics in the vineyard using ground and aerial survey methods. The paper reports results for two years of the study.
The article has the following deficiencies:
1. The purpose of the study is stated too broadly. The authors should clarify and specify the purpose of the study. After, perhaps, expand the methodology of the study.
2. The abstract should specify the purpose, methodology of the study, expand the results by indicating the obtained numerical values.
3. The introduction should expand on the literature review of methods, techniques, and vegetative indices for soil analysis in vineyards.
4. According to the technical documentation, the GSD of the Parrot Sequoia at an altitude of 120 meters is 14.8 cm/pixel. The authors indicate 0.12 m/pixel. Explain the discrepancy.
5. Specify the main flight parameters: speed, overlap.
6. The results are different by year (Lines 650-655). The authors draw conclusions about the differences. Perhaps the result was influenced by other factors, random coincidences. What do the studies of other authors say about this? Perhaps it is necessary to analyze the data for the third year. Explanations should be added to the Discussion section.
Other comments:
1. Line 63 is an extra parenthesis and period
2. line 121. Figure 4a is not easily located for viewing. Figures should be numbered as they are mentioned in the text.
3. lines 559-561 Text delete (Text from template)
Author Response
The authors conducted a study of soil characteristics in the vineyard using ground and aerial survey methods. The paper reports results for two years of the study.
Thank you very much for your suggestions. We have corrected in the manuscript the issues stated by the reviewer. All the changes are in yellow in the new version of the manuscript.
The article has the following deficiencies:
- The purpose of the study is stated too broadly. The authors should clarify and specify the purpose of the study. After, perhaps, expand the methodology of the study.
The purpose of the study was described at the end of the introduction.
- The abstract should specify the purpose, methodology of the study, expand the results by indicating the obtained numerical values.
Done
3.The introduction should expand on the literature review of methods, techniques, and vegetative indices for soil analysis in vineyards.
Done
- According to the technical documentation, the GSD of the Parrot Sequoia at an altitude of 120 meters is 14.8 cm/pixel. The authors indicate 0.12 m/pixel. Explain the discrepancy.
The information has been corrected.
- Specify the main flight parameters: speed, overlap.
The information has been included in the methodology “UAVs images”
- The results are different by year (Lines 650-655). The authors draw conclusions about the differences. Perhaps the result was influenced by other factors, random coincidences. What do the studies of other authors say about this? Perhaps it is necessary to analyze the data for the third year. Explanations should be added to the Discussion section.
Done
Other comments
- Line 63 is an extra parenthesis and period.
Done
- line 121. Figure 4a is not easily located for viewing. Figures should be numbered as they are mentioned in the text.
The figure has been moved to ensure its explanation corresponds with the text. Now, the figure corresponds to the numbering 3a.
- lines 559-561 Text delete (Text from template)
Done
Thank you for your valuable feedback. We sincerely appreciate your input and hope the updated version meets your expectations.
Round 2
Reviewer 1 Report
The author has revised the manuscript, but the revisions still fail to address the previously raised issues. For example, Tables 2-4 does not provide the number of observations. Meanwhile, I strongly recommend polishing the manuscript before publication.
Extensive editing of English language required
Author Response
Dear revisor,
Thanks to the comments to the document.
The modifications have been highlighted on the manuscript.
I enclose the response to the comments of the reviewers.
Much regards
Luz
The author has revised the manuscript, but the revisions still fail to address the previously raised issues. For example, Tables 2-4 does not provide the number of observations.
Three repetitions of each sample for each horizon and soil trench were analysed.
Table 2-4 descriptions were modified with this information.
Meanwhile, I strongly recommend polishing the manuscript before publication.
Done
Does the introduction provide sufficient background and include all relevant references? Can be improved.
The introduction has been modified to include additional background.
Are all the cited references relevant to the research? Can be improved
Relevant information has been included.
Is the research design appropriate? Can be improved
The description of the research design was improved.
Are the methods adequately described? Can be improved
The methodology of soil description was improved.
Are the results clearly presented? Can be improved.
The tables of soil description were improved.
Are the conclusions supported by the results? Must be improved.
The results and discussion have been modified to support conclusions.
Extensive editing of English language required.
MPDI language editing services have revised the document.
Reviewer 2 Report
The article is revised and I recommend to accept in its present from.
The article is revised and I recommend to accept in its present from.
Author Response
Dear revisor,
Thanks to the comments to the document.
The modifications have been highlighted on the manuscript.
I enclose the response to the comments of the reviewers.
Much regards
Luz
The article is revised and I recommend to accept in its present from.
Does the introduction provide sufficient background and include all relevant references? Can be improved.
The introduction has been modified to include additional background.
Are all the cited references relevant to the research? Can be improved
Relevant information has been included.
Extensive editing of English language required.
MPDI language editing services have revised the document.
Reviewer 3 Report
The authors have taken into account the reviewers' comments on the finalization of the article. At the same time, the authors should pay attention to the display of figures and tables, for example, Table 10, as well as check the text for the quality of the English language.
Check the text for English quality, especially prepositions.
Author Response
Dear revisor,
Thanks to the comments to the document.
The modifications have been highlighted on the manuscript.
I enclose the response to the comments of the reviewers.
Much regards
Luz
The authors have taken into account the reviewers' comments on the finalisation of the article. At the same time, the authors should pay attention to the display of figures and tables, for example, Table 10, as well as check the text for the quality of the English language.
The format of tables have been corrected.
Extensive editing of English language required.
MPDI language editing services have revised the document.